# Co-Delivery of Novel Synthetic TLR4 and TLR7/8 Ligands Adsorbed to Aluminum Salts Promotes Th1-Mediated Immunity against Poorly Immunogenic SARS-CoV-2 RBD

**DOI:** 10.3390/vaccines12010021

**Published:** 2023-12-23

**Authors:** Karthik Siram, Stephanie K. Lathrop, Walid M. Abdelwahab, Rebekah Tee, Clara J. Davison, Haley A. Partlow, Jay T. Evans, David J. Burkhart

**Affiliations:** Center for Translational Medicine, Department of Biomedical and Pharmaceutical Sciences, University of Montana, Missoula, MT 59812, USA; karthik.siram@mso.umt.edu (K.S.); stephanie.lathrop@mso.umt.edu (S.K.L.); walid.abdelwahab@mso.umt.edu (W.M.A.); rebekah.tee@mso.umt.edu (R.T.); clara.davison@mso.umt.edu (C.J.D.); haley.partlow@umconnect.umt.edu (H.A.P.); jay.evans@mso.umt.edu (J.T.E.)

**Keywords:** toll-like receptor (TLR) agonist, TLR4, TLR7/8, adjuvants, alhydrogel, adju-phos, SARS-CoV-2, COVID-19

## Abstract

Despite the availability of effective vaccines against COVID-19, severe acute respiratory syndrome coronavirus 2 (SARS-CoV-2) continues to spread worldwide, pressing the need for new vaccines with improved breadth and durability. We developed an adjuvanted subunit vaccine against SARS-CoV-2 using the recombinant receptor–binding domain (RBD) of spikes with synthetic adjuvants targeting TLR7/8 (INI-4001) and TLR4 (INI-2002), co-delivered with aluminum hydroxide (AH) or aluminum phosphate (AP). The formulations were characterized for the quantities of RBD, INI-4001, and INI-2002 adsorbed onto the respective aluminum salts. Results indicated that at pH 6, the uncharged RBD (5.73 ± 4.2 mV) did not efficiently adsorb to the positively charged AH (22.68 ± 7.01 mV), whereas it adsorbed efficiently to the negatively charged AP (−31.87 ± 0.33 mV). Alternatively, pre-adsorption of the TLR ligands to AH converted it to a negatively charged particle, allowing for the efficient adsorption of RBD. RBD could also be directly adsorbed to AH at a pH of 8.1, which changed the charge of the RBD to negative. INI-4001 and INI-2002 efficiently to AH. Following vaccination in C57BL/6 mice, both aluminum salts promoted Th2-mediated immunity when used as the sole adjuvant. Co-delivery with TLR4 and/or TLR7/8 ligands efficiently promoted a switch to Th1-mediated immunity instead. Measurements of viral neutralization by serum antibodies demonstrated that the addition of TLR ligands to alum also greatly improved the neutralizing antibody response. These results indicate that the addition of a TLR7/8 and/or TLR4 agonist to a subunit vaccine containing RBD antigen and alum is a promising strategy for driving a Th1 response and neutralizing antibody titers targeting SARS-CoV-2.

## 1. Introduction

The current severe acute respiratory syndrome coronavirus 2 (SARS-CoV-2) pandemic has been ongoing for more than two years and continues to challenge the available science and technology worldwide to combat COVID-19 infection. Globally, as of 8 November 2023, there were 771,820,937 confirmed cases of COVID-19, including 6,978,175 deaths deaths, as reported to the WHO [1]. The prevailing SARS-CoV-2 vaccines in the market have been developed using either mRNA technology [2,3], adenovirus vectors [4], whole inactivated virus [5,6], or as a protein-based subunit vaccine [7]. Although the existing vaccines have successfully elicited a robust immune response, this success is hampered by the hurdles of a slow supply chain, the need for storage at extremely low temperatures, and high cost. These factors limit the use of the existing vaccines across all the nations in the world.

SARS-CoV-2 is an enveloped virus that depends on the spike (S) glycoprotein for binding and entry of host cells. The S protein exists as a homotrimer on the viral envelope, where binding of the receptor–binding domain (RBD) to human angiotensin-converting enzyme 2 (hACE2) initiates cell entry. Antibodies targeting spike, particularly its RBD, can efficiently neutralize the virus; therefore, it is the primary target for neutralizing antibodies in the current vaccines [8]. Due to the complexity of producing the large protein trimer, there has been much interest in using the recombinantly expressed RBD portion of the spike as a vaccine target [9].

Although there have been several efforts to develop RBD-based subunit vaccines, these have been hindered due to the poor immunogenicity of the RBD protein component [9]. In a subunit vaccine, an adjuvant is often required to engage the innate immune system due to the absence of the pathogen-associated molecular patterns (PAMPs) that would be present in natural infection. Additionally, adjuvants can help reduce the needed dose of antigen, which is of great help during a pandemic when large-scale vaccinations are required. This was demonstrated during the 2009 flu pandemic, where H1N1 vaccines adjuvanted with adjuvant system 03 (AS03) generated higher antibody titers compared with the non-adjuvanted vaccines that contained a higher dose of antigen [10]. Additionally, subunit vaccines are relatively easy to manufacture, inexpensive, highly effective, generally do not require stringent storage conditions, and have a good safety profile.

Selection of the adjuvant is critical, as it allows for the tailoring of the immune response to be most effective against the specific pathogen [11,12]. Examples of adjuvants approved for human use in licensed vaccines include Algel-Imidazoquinoline [13], alhydrogel [14], CpG 1018 [14], AS01 [15,16], AS03 [10,16], AS04 [15,17], MF59 [15,17], and Matrix-M [7]. Most approved adjuvants activate either individual or combinations of pathogen recognition receptors (PRRs) to drive stronger humoral and/or cell-mediated immunity [11,15,18,19]. One well-studied class of PRRs is toll-like receptors (TLR), which comprise an evolutionarily conserved receptor family capable of detecting and responding to various microbial challenges [20,21].

Human TLR7 receptors are present within plasmacytoid dendritic cells and B cell endosomes, while TLR8 is found more widely distributed in myeloid cells [22]. These receptors both recognize single-stranded RNA, as well as a number of synthetic oxoadenines and imidazoquinolones. These TLRs signal through myeloid differentiation primary response gene 88 (MyD88) to produce type I interferons and the pro-inflammatory cytokines TNFα, IL-1β, MIP-1α, IL-2, IL-6, and IL-12 [20]. INI-4001 is a novel lipidated TLR7/8 ligand with an oxoadenine core (Figure 1A), designed to signal through both TLR8 and TLR7 effectively and easily formulated in liposomes, emulsions, or by adsorption to alum with reduced side effects previously associated with the rapid systemic distribution of core TLR7/8 ligands [23]. The success of subunit vaccines that employ TLR agonists is demonstrated by Heplisav-B™ (Dynavax, Emeryville, CA, USA), Shingrix^®^, Mosquirix^®^, Fendrix^®^, and Cervarix^®^ (GlaxoSmithKline, Brentford, UK) [16].

Human TLR4 receptors are present on the cell surface of antigen-presenting cells and recognize bacterial cell wall components such as lipopolysaccharides. Their activation produces multiple inflammatory cytokines, including TNFα, IL-1β, IL-6, IL-8, and IL-12, through the initiation of MyD88 and TIR domain-containing adaptor protein inducing interferon-beta (TRIF)-dependent pathways [21,24]. INI-2002 is a novel synthetic TLR4 ligand derivatized from lipopolysaccharide (Figure 1B). The promise of using a TLR4 agonist as a vaccine adjuvant has been demonstrated by the approval of GlaxoSmithKline’s Adjuvant System 04, an adjuvant consisting of the TLR4 agonist MPL (3-O-desacyl4′-monophosphoryl lipid A) adsorbed onto a particulate form of aluminum salt. It is licensed for use in two human vaccines, Cervarix^®^ and Fendrix^®^ (GlaxoSmithKline, Brentford, UK), and was shown not only to boost the antigen-specific antibody response but also to promote epitope broadening [25,26].

For decades, aluminum salts have been used successfully as adjuvants in subunit vaccines, boosting antibody titers and promoting a Th2-mediated immune response through adsorption to antigens [11,27]. Aluminum salts have many benefits that make them ideal adjuvants for a pandemic vaccine: a strong safety profile, a long history of regulatory approval, low cost, good stability, and ease of manufacture, storage, and distribution. Alhydrogel (AH) and Adju-Phos (AP) are both aluminum-containing salts, composed of a semi-crystalline form of aluminum oxyhydroxide and amorphous salt of aluminum hydroxyphosphate, respectively [27].

We made several aluminum salt formulations that can efficiently adsorb and co-deliver TLR7/8 agonist INI-4001 and/or TLR4 agonist INI-2002 with Spike RBD antigen. Our lab has previously demonstrated that co-delivery of TLR7/8 and TLR4 ligands within a liposome increased IgG2a antibody titers and skewed the immune response to an influenza antigen towards Th1 [28]. Additionally, we showed that TLR agonists, INI-4001 and INI-2002, could similarly enhance and skew the immune response from Th2 to Th1-dominated when added to an emulsion-based adjuvant in the context of vaccination against SARS-CoV-2 spike RBD [29]. In line with these results, we developed subunit SARS-CoV-2 vaccines using RBD as the antigen, co-delivered with an aluminum salt alone or in combination with a TLR7/8 agonist (INI-4001) and/or a TLR4 agonist (INI-2002). These vaccine formulations were tested in C57BL/6 mice for their ability to enhance immunity to RBD. We measured the anti-RBD serum antibody titers and also evaluated their ability to block the binding of spike RBD and the full-length spike trimer to hACE2. Additionally, we measured the cytokines produced by antigen-specific T cells to determine the profile of the cellular response. We found that co-delivery of RBD with INI-4001 and/or INI-2002 on AH efficiently skewed the immune response towards Th1 and generated strong neutralizing antibodies.

## 2. Materials and Methods

### 2.1. Antigens and Adjuvants

The compound 2-[(R)-3-decanoyloxytetradecanoylamino]ethyl 2,3-di-[(R)-3-decanoyloxytetradecanoylamino]-2,3-dideoxy-4-O-sulfoxy-β-D-allopyranoside (INI-2002) was synthesized as previously described [30]. UM-4001 was synthesized by the phospholipidation of 6-amino-2-butoxy-9-[(1-hydroxyethyl-4-piperidinyl)-methyl]-7, 9-dihydro-8H-purin-8-one [31] using the published phosphoramidite method [32]. AH (catalog code vac-alu-250) and AP (catalog code vac-phos-250) were purchased from InvivoGen (San Diego, CA, USA). RBD antigen was synthesized by expression in HEK-293 cells and subsequent purification [33].

### 2.2. Adsorption of INI-4001, INI-2002, and RBD to Preformed Aluminum Salts

Aqueous formulations of INI-4001 and INI-2002 at 2 mg/mL were prepared by solubilizing the respective ligand in 2% glycerin using a bath sonicator (FB11201, Fisherbrand, Thermo Fisher Scientific, Waltham, MA, USA) for 3 h or 15 min, respectively, at a temperature < 35 °C. The formulations were sonicated until the particle size was below 200 nm. The particle size was measured using Zetasizer Nano-ZS (Malvern Panalytical, Malvern, UK) after 10X dilution with WFI. The aqueous formulations were sterile-filtered using a 13 mm Millex GV PVDF filter with a pore size of 0.22 μm (MilliporeSigma, Burlington, MA, USA).

Prior to the adsorption experiments, AH and AP were diluted to 1 mg/mL using the corresponding formulation vehicle. Adsorption experiments of INI-4001, INI-2002, and RBD to AH were performed using two formulation vehicles, 2% glycerin or 10 mM TRIS buffer of pH 8.1 (referred to as TRIS buffer hereafter). Adsorption experiments of INI-4001, INI-2002, and RBD to AP were performed using 2% glycerin. The weight ratio of aluminum salt (AH or AP) to the sum of antigen and adjuvant (INI-4001 and/or INI-2002) was 2:1. A series of formulations were prepared by mixing the required amounts of INI-4001, INI-2002, RBD, AH, AP, and formulation vehicle by end-over-end rotation at room temperature for 1 h [34]. The compositions of the formulations are described in Table 1.

### 2.3. Measurement of Zeta Potential and pH of the Adsorbed Formulations

Zeta potential was measured using Zetasizer Nano-ZS (Malvern Panalytical, Malvern, UK) by a disposable capillary cell following the manufacturer’s instructions. A 1:10 dilution in the respective formulation vehicle was used for all the samples. The pH of the vaccine formulations was measured using an Accumet AB150 pH meter (Thermo Fisher Scientific, Waltham, MA, USA) attached with an InLab Microprobe (Mettler-Toledo, Columbus, OH, USA) after a three-point calibration using pH 4.01, 7.00, and 10.01 standards. All the sample measurements were performed in triplicate.

### 2.4. Visualization of AH and AP Formulations by Cryo-TEM

The morphology of the formulations was visualized using a Thermo Fisher Scientific Glacios cryo-TEM at 200 kV with a K3 Gatan direct electron detector. AH and AP formulations adsorbed with INI-4001, INI-2002, and RBD were vortexed for 2 min, and 3 µL of the samples were pipetted onto glow-discharged (120 s 15 mAmp, negative mode) copper Quantifoil holey carbon support grids (Ted Pella 658-300-CU) and vitrified in liquid ethane using a Mark IV Vitrobot (Thermo Fisher, Hillsboro, OR, USA). The conditions used for the cryopreservation were 100% humidity, blot force −15, and blotting time 3 s. Cryo-EM images were collected with a defocus range of 2.5 µm.

### 2.5. Analytical Method to Simultaneously Analyze INI-4001 and INI-2002

Simultaneous quantitation of INI-4001 and INI-2002 was performed using a Waters Acquity Arc RP-HPLC system equipped with a 2998 PDA detector. Separation was performed using a Waters CORTECS C18^+^ 3.0 × 50 mm 2.7 µm column at 40 °C. Mobile phases comprising 10 mM ammonium formate at pH 3.2 with acetonitrile and water (60:40 ratio) were used as mobile phase A, and 10 mM ammonium formate at pH 3.2 with acetonitrile and IPA (50:50 ratio) was used as mobile phase B. The gradient (0–0.5 min 60% A; 6–7 min 100% B; 7.1–8.5 min 60% A) was run for 8.5 min at 1.5 mL/min. The absorbance of INI-4001 and INI-2002 was measured at 278 nm and 210 nm, respectively. All the samples were quantitated by peak area based on extrapolation from a six-point dilution series of the corresponding standards in methanol.

### 2.6. Method to Determine Free INI-4001, INI-2002, and RBD with Aluminum Salts

To detect the unbound INI-4001, INI-2002, and RBD on the aluminum salt formulations, 100 µL of the formulation was centrifuged for 5 min at 19,400 rcf (5417C, Eppendorf) to form sediment of aluminum salt at the bottom. The supernatants were carefully collected and analyzed for the amount of free INI-4001, INI-2002, and RBD. The amount of INI-4001 and INI-2002 in the supernatant samples was analyzed using the RP-HPLC method mentioned above. A Coomassie plus kit (ThermoFisher Scientific, product number 23236) was used by following the manufacturer’s protocol to quantify the amount of RBD in the supernatants. The percentage of RBD adsorbed on AH and AP was estimated relative to RBD recovered in the control samples.

### 2.7. In Vivo Studies

Eight- to ten-week-old C57BL/6 mice (Jackson Laboratory, Bar Harbor, ME, USA) were group housed under a 12/12 h light/dark cycle and given food and water ad libitum. Animal studies were carried out in an OLAW and AAALAC-accredited vivarium in accordance with the University of Montana IACUC guidelines (protocol 015-19JEDBS-041919). Mice were vaccinated with 50 µL in one gastrocnemius (calf) muscle using a 27G needle, with 8 mice per experimental group and 5 mice in the vehicle-only control group. Vaccines consisted of 1 µg of recombinant SARS-CoV-2 RBD, along with 24 µg per mouse of Alhydrogel (Invivogen, San Diego, CA, USA) or Adju-phos (Invivogen, San Diego, CA, USA) and/or 10 µg of the TLR4 agonist INI-2002 (Inimmune Corp, Missoula, MT, USA), 10 µg of the TLR7/8 agonist INI-4001 (Inimmune Corp, Missoula, MT, USA) or both, brought up to volume with 2% glycerol/water. These doses of INI-2002 and INI-4001 were chosen based on previous experiments that showed good adjuvant activity without notable reactogenicity at these doses [29], and no sign of these was noted in the current experiments. Mice were given an identical booster vaccine 14 days later, and blood was sampled by submandibular collection 28 days after the booster. Blood samples were allowed to coagulate in BD Microtainer serum separator tubes, and the serum was collected following a 5 min centrifugation at approximately 16,000 rcf and stored at −20 °C until assayed. A second booster was administered on day 42 (28 days post-secondary), and mice were sacrificed 5 days later to assess the functional phenotype of RBD-specific T cells. The draining (inguinal and popliteal) LNs from the side of the injection were collected in cold HBSS. Organs were manually disassociated into single-cell suspensions, resuspended in cold HBSS, and counted using a Cellaca MX (Nexelcom Biosciences). The cells were resuspended to the desired concentration in complete RPMI (RPMI-1640, Gibco/ThermoFisher) containing 2 mM L-glutamine, 100 U/mL penicillin, 100 U/mL streptomycin (Cytivia/Hyclone), 550 µM b-mercaptoethanol (Gibco/Thermo Fisher), and 10% heat-inactivated fetal bovine serum (Cytivia/Hyclone)). 2 × 10^6^ LN cells from individual mice were cultured in flat-bottom 96-well TC-treated plates for approximately 72 h in the presence of 10 µg/mL RBD protein. The plates were centrifuged for 5 min at 400× *g*, and the cell supernatant was recovered and stored at -20 °C until assayed for cytokines.

### 2.8. Serum ELISAs for Quantification of Anti-RBD Antibodies

Serum samples were analyzed using ELISA for RBD-specific IgG, IgG1, and IgG2c antibody titers. Nunc MaxiSorp 96-well plates (Thermo Fisher Scientific) were coated overnight at RT with 100 µL per well of RBD antigen at 5 µg/mL (for IgG) or 2.5 µg/mL (for IgG1 and IgG2c) in PBS. The plates were washed three times with PBS + 0.5% Tween-20 (PBS-T) and blocked with 200 µL of SuperBlock (ScyTek Laboratories, Logan, UT, USA) for 1 h at 37 °C. Serial dilutions of each serum sample were made in EIA buffer (1% bovine serum albumin, 0.1% Tween-20, and 5% heat-inactivated fetal bovine serum in phosphate-buffered saline) according to the expected antibody response. The blocking solution was removed, and plates were incubated with diluted serum for 2 h at 37 °C, followed by washing three times with PBS-T and incubation with anti-mouse IgG-HRP (1:3000), IgG1-HRP (1:5000) or IgG2c-HRP (1:5000) secondary antibody (SouthernBiotech, Birmingham, AL, USA) in EIA buffer for 1 h. After three washes, secondary antibodies were detected by the addition of the TMB Substrate (KPL SureBlue™, SeraCare, Milford, MA, USA) for 20 min, followed by stopping the reaction by the addition of H_2_SO_4_. The SpectraMax 190 microplate reader (Molecular Devices, San Jose, CA, USA) was used to measure absorbance at 450 nm, and the antibody titers are reported as the dilution factor required to achieve an optical density of 0.3. Measurements below the limit of detection for sera diluted at 1:50 were assigned a value of 1.0 for analysis purposes.

### 2.9. T-Cell Cytokine Assessment

The concentrations of IFNγ, IL-5, and IL-17A in the supernatants collected from spleen and LN cells isolated from individual animals following stimulation with RBD for 72 h was assessed using a U-Plex assay (multiplex ELISA) by MesoScale Discovery (MSD) according to the manufacturer’s protocol.

### 2.10. Surrogate Neutralization Assay

Serum samples were assayed for the ability to inhibit the binding of soluble human ACE2 to the SARS-CoV-2 receptor–in binding domain (RBD) and the trimerized protein ectodomain (Spike) as a surrogate for viral neutralization capability. Sera from individual mice were assayed with the V-PLEX SARS-CoV-2 ACE2 Panel 6 Kit (K15436U, Meso Scale Diagnostics, Rockville, MD, USA) using the supplied standard antibody and according to the manufacturer’s instructions. Briefly, this kit contains RBD and Spike proteins immobilized to plates. The detection of bound human ACE2 protein conjugated to SULFO-TAG labels is conducted by electrochemiluminescence detection using a MESO QuickPlex SQ 120 imager (Meso Scale Diagnostics). A standard monoclonal Ab that binds to RBD and efficiently blocks its binding to human ACE2 was used as a standard, and these sample data are reported as the µg/mL concentration of the standard antibody that gave equivalent blocking of binding through regression analysis.

### 2.11. Statistics

RBD-specific antibody titers, antigen-specific cytokine concentrations, and surrogate neutralization assay results were plotted, and the statistics were calculated using Prism (GraphPad Software v.10, LLC). For RBD-specific antibody titers and surrogate neutralization assays, significant differences between groups were determined using a Brown–Forsythe and Welch ANOVA of the log-transformed data with Dunnett’s T3 multiple comparisons test. For cytokine concentrations, a one-way ANOVA was performed on these log-transformed data with a Tukey post-test. Asterisks indicate statistical differences between relevant groups; * = *p* < 0.05, ** = *p* < 0.01, *** = *p* < 0.001, **** = *p* < 0.0001.

## 3. Results and Discussion

The long history of alum as a safe and successful vaccine adjuvant caused us to hypothesize it would synergize with TLR-based adjuvants to improve the immune response against the weakly immunogenic SARS-CoV-2 spike RBD. The adjuvant activity of alum is thought to depend, at least in part, on the adsorption of the antigen to the aluminum salts [35]. Therefore, we sought to adsorb not only the antigen but also the TLR ligands to the alum in order to maximize the co-delivery to relevant antigen-presenting cells.

### 3.1. Pre-Formulation Studies

Pre-formulation studies were performed to optimize the amount of aluminum salt (AH or AP) needed to fully adsorb INI-4001, INI-2002, or RBD, as well as the duration of mixing required to achieve complete adsorption. The results indicated efficient adsorption of INI-4001, INI-2002, or RBD to aluminum salt at a 1:2 weight ratio (Appendix A). Hence, this weight ratio was used for all the subsequent formulation studies reported herein. Although complete adsorption was achieved in 30 min, the duration of mixing was increased to 60 min to ensure complete adsorption.

### 3.2. Adsorption Characteristics of INI-4001, INI-2002, and RBD to AH

The adsorption results of the TLR ligands INI-4001 and INI-2002 to AH in Figure 2 indicated efficient adsorption in 2% glycerin or TRIS buffer. The results also demonstrate that neither TLR ligand desorbed when used in combination nor the presence of RBD. Adjuvants and antigens adsorb to AH either by ligand exchange, electrostatic attraction, or hydrophobic forces [17]. As shown in Figure 1, INI-4001 and INI-2002 have phosphate and sulfate groups, respectively, in their structures that could facilitate ligand exchange with the hydroxyl groups of AH to promote adsorption. The zeta potential values (Figure 3) of INI-4001 and INI-2002 in 2% glycerin and TRIS buffer are negative, whereas the zeta potential of AH in 2% glycerin and TRIS buffer is positive. After mixing INI-4001 with AH, the zeta potential of AH was reduced from 25 mV to 11.6 ± 2.6 (2% glycerin) and 6.8 ± 0.1 mV (TRIS), indicative of adsorption. Similarly, the mixing of INI-2002 to AH decreased the zeta potential of AH to 9.4 ± 4.1 (in 2% glycerin) and 5.9 ± 0.1 mV (in TRIS buffer). Therefore, adsorption may also occur by electrostatic attraction.

The adsorption results for RBD to AH in Figure 2A indicated that only 6% of RBD adsorbed to AH at a weight ratio of 1:2 when 2% glycerin (pH 6) was used as the formulation vehicle. The adsorption efficiency of RBD to AH did not increase even when using 24 times the amount of AH. This suggests that RBD does not adsorb to AH. However, the adsorption efficiency of RBD to AH at 1:2 significantly improved to 80% (*p*-value = 0.0002) when TRIS buffer at pH 8.1 was used as the formulation vehicle. The isoelectric point of RBD is 7.78, indicating RBD is positively charged at a pH lower than 7.78 and negatively charged at a pH above 7.78. When 2% glycerin was used as the formulation vehicle, the zeta potential of RBD and AH was +5.7 mV (pH 6.09) and +22.68 mV (pH 6.36), respectively (Figure 3B). The zeta potential of AH did not change after mixing with RBD, which corroborates the fact that the neutrally charged RBD does not adsorb to the positively charged AH. When TRIS buffer at pH 8.1 was used as the formulation vehicle, the zeta potential of RBD flipped to a negative charge (−18.20 mV at pH 7.8), which facilitated its adsorption to positively charged AH by electrostatic attraction (Figure 3C). Hence, switching the formulation vehicle from 2% glycerin to TRIS buffer greatly increased the adsorption efficiency. This approach of modulating the formulation pH in order to flip the charge of the antigen and thereby enhance its adsorption can be used for cationic antigens that do not otherwise adsorb to AH.

Alternately, RBD efficiently adsorbed to AH without TRIS buffer when the AH was pre-adsorbed to INI-4001 and/or INI-2002, improving from 6.5% to more than 65% (Figure 2A). With the addition of RBD, the zeta potential of the AH formulation with INI-4001 changed from +11 mV to +15 mV (Figure 3B), suggestive of adsorption of RBD to AH. The adsorption of anionic ligands on the surface of the cationic AH could provide negatively charged areas to facilitate the subsequent adsorption of RBD. Thus, the combination of INI-4001 and/or INI-2002, along with RBD, can be adsorbed in the correct order of addition and co-delivered with AH, even at pH 6.

### 3.3. Adsorption Characteristics of INI-4001, INI-2002, and RBD to AP

Although we could achieve efficient adsorption of TLR ligands and RBD to AH, we also evaluated the adsorption of TLR ligands and RBD to AP as a potential alternative. INI-4001 and INI-2002 both adsorbed poorly to AP (Figure 4B,C), with an adsorption efficiency of approximately 20%. Unlike AH, AP does not possess the hydroxyl groups necessary for ligand exchange with the functional groups in an adjuvant. Additionally, the aqueous formulations of INI-4001 and INI-2002 are negatively charged (−31 mV at pH 6.9), making it more challenging for the TLR ligands to adsorb to negatively charged AP.

On the other hand, RBD adsorbed efficiently to the negatively charged AP in 2% glycerin with an adsorption efficiency of 87% (Figure 4A). After mixing with RBD, the zeta potential of AP changed from −31 mV (pH 6.9) to + 3 mV (pH 6.6) (Figure 3A), indicative of the adsorption of RBD to the negatively charged AP. Interestingly, the adsorption of RBD on AP promoted the subsequent adsorption of the TLR ligands. As observed in Figure 4B,C, INI-4001 and INI-2002 adsorbed poorly to AP in the absence of RBD, but their adsorption efficiencies increased with the inclusion of RBD. These results suggested an enhancement in the adsorption of the TLR ligands to AP in the presence of RDB. Thus, both the TLR ligands (INI-4001 and INI-2002) and RBD can be adsorbed and co-delivered using AP, albeit to a lower level than the optimized AH formulation.

### 3.4. Adsorption Characteristics of INI-4001, INI-2002, and RBD to a Mixture of AH and AP

In light of the fact that the TLR ligands and RBD efficiently adsorbed to AH and AP, respectively, we also developed a formulation containing both the TLR ligands pre-adsorbed to AH (INI-4001 and INI-2002) and RBD pre-adsorbed to AP. Consistent with the previous results, RBD and the TLR ligands adsorbed well to AH and AP, respectively (Figure 5A). Upon mixing, the TLR ligands did not desorb (Figure 5B), but the addition of all the components lowered the adsorption efficiency of RBD from 90% to 53%, indicating the desorption of RBD from AP. The zeta potential of the formulation composed of only AH and AP was +16.17 mV (pH 5.95), while that of the formulation comprising AP, AH, and TLR agonists was +4.47 mV (pH 6.6). This reduction indicates the adsorption of the TLR agonists to AH. The addition of RBD to the formulation of AP, AH, and TLR agonists increased the zeta potential to 6.87 (pH 6.53), reflecting the partial adsorption of RBD in the formulation.

### 3.5. Visualization of AH and AF Formulations

The morphology of AH and AP formulations adsorbed with INI-4001, INI-2002, and RBD were visualized using cryo-TEM microscopy. Consistent with previously published results, both formulations appeared to have particles with dimensions around 100 nm [34,36,37]. The images of AH formulation adsorbed with TLR agonists, and RBD showed long and crystalline AH particles adsorbed with nanoparticles of TLR ligands (Figure 6A). In contrast, AP particles in Figure 6B appeared to have amorphous quasi-shaped structures. Similar structures for AP have been previously reported earlier [38,39].

### 3.6. In Vivo Evaluations of Alum Adsorbed Formulations

To determine the impact of adsorption and co-delivery of TLR ligands and RBD to aluminum salts on vaccine-mediated immunity, we vaccinated C57BL/6 mice with different combinations of TLR ligands and RBD adsorbed to AH and AP. Mice were dosed with 1 µg RBD, with 10 µg INI-4001, and/or 1/10 µg INI-2002. The amount of aluminum salt (AH or AP) used for dosing was two times the mass of antigen and adjuvant (INI-4001 or INI-2002) in each formulation.

### 3.7. The Addition of TLR Ligands to Alum-Adjuvanted RBD Enhances the Antibody Response

Mice were vaccinated twice, 14 days apart, with 1 µg of purified, recombinant spike RBD with or without the indicated adjuvants (Figure 7A). Serum was collected 28 days after the second vaccination, and serum anti-RBD antibody titers were measured using ELISA. As shown in Figure 7B (far left), animals that received RBD antigen alone (filled black circles) exhibited very low RBD-specific antibody titers, similar to control mice injected with vehicle alone (open black circles). In the absence of alum, the addition of TLR4 ligand INI-2002 (red) significantly increased RBD-specific IgG titers, while the TLR7/8 ligand INI-4001 (blue) gave only a modest boost (not statistically significant). However, when the two TLR agonists were combined, an even greater increase in anti-RBD titer was measured (purple triangles, *p* < 0.0001). Significant increases were noted in both IgG2c (Figure 7C) and IgG1 (Figure 7D), but the IgG2c levels were preferentially increased, suggestive of a Th1-dominated response (Figure 7E). As both TLR agonists are negatively charged, it seems likely that they interact with the positively charged RBD antigen, facilitating co-delivery to antigen-presenting cells in order to promote the vaccine-induced immune response.

Adjuvanting RBD with AH (either at pH 6 or 8.1) or AP resulted in modestly increased IgG titers (green symbols). No significant difference in antibody titers between AH used at pH 6 or 8.1 was seen, although a trend towards higher titers was seen at pH 8.1 (the TRIS formulation with greatly improved RBD adsorption; Figure 2A). Additionally, despite the high adsorption of RBD by AP (Figure 3A), this form of alum provided the lowest boost in antibody titer. These observations suggest that the level of adsorption to alum, as measured in the laboratory, cannot be used to predict the efficacy of adjuvantation. It is possible that these adsorption measurements are not representative of conditions in vivo.

The addition of either TLR ligand to alum significantly enhanced the anti-RBD titers beyond the level seen with either alum formulation alone. The relative increase in IgG1 tended to be similar to or higher than IgG2c for most formulations containing AH (Figure 7E), reflective of the tendency of alum to promote a Th2 response. Interestingly, the AP-containing formulations did not follow this trend. For vaccines containing both TLR ligands in combination, the presence of alum did not significantly augment the response above the level in the absence of alum (compare purple symbols). However, the presence of alum served to greatly increase the antibody titers from INI-4001 (compare blue symbols).

When speculating on the mechanisms by which these adjuvants interact with RBD and each other, we can imagine differences in both the likelihood of antigen uptake by appropriate antigen-presenting cells (APC) and the ability of the TLR ligands to interact with their receptors. We hypothesize that the negatively charged TLR agonists would directly interact with the positively charged RBD antigen (Figure 8, far left). In this situation, the TLR4 ligand INI-2002 will be surface accessible and capable of engaging TLR4 receptors on antigen-presenting cells to promote the desired immune response. However, in the absence of the TLR4 ligand, the TLR7/8 agonist does not enhance immunogenicity nearly as well, perhaps due to the dependence of the TLR7/8 agonist on cellular uptake to engage intracellular receptors.

The reduced activity of the TLR7/8 ligand INI-4001 is largely overcome in the presence of either type of alum. As alum has been shown to promote the recruitment of dendritic cells and macrophages to the draining lymph nodes [40,41,42,43,44], we hypothesize it will perform the same function when combined with the TLR agonists, thereby increasing the likelihood of antigen and adjuvant uptake. Additionally, alum will likely increase the particle size by interacting with the antigen and TLR agonist, serving to further promote uptake by antigen-presenting cells (Figure 8, center and right). By promoting cellular uptake, it also allows INI-4001 to engage its intracellular receptor to improve the immune response further. This schematic helps us imagine how the combination of one or both TLR ligands and either form of alum could serve to promote the immune response by facilitating the adsorption and formation of larger particles. Indeed, the antibody titers from all TLR ligand/alum combinations are significantly increased over that of antigen alone and, in most cases, over alum alone.

### 3.8. Cell-Mediated Immunity to RBD Is Modulated by Alum and TLR Ligands

Relative antibody isotype titers, as shown in Figure 7, indicate the type of CD4^+^ T helper response that has developed, but we also sought a more direct measure of the antigen-specific T cell phenotype that developed in lymphoid tissues following vaccination. Single-cell suspensions isolated from the draining lymph nodes (dLN) of animals five days after a 3rd vaccination were cultured with RBD for approximately 72 h to restimulate existing antigen-specific T cells. Cytokines released into the cell supernatant during this restimulation were measured using multiplex ELISA (Figure 9, Appendix A). IFNγ is a signature cytokine released by Th1 T cells and was used as an indication of Th1 development. Similarly, IL-17A is released by Th17 cells, and IL-5 is a signature cytokine secreted by Th2 cells (along with IL-4 and IL-13, not measured) and was used as an indication of these cell types.

In the absence of alum (triangles, left side), the TLR7/8 ligand INI-4001 (blue) induced a Th1-biased response, as judged by high IFNγ production and very little IL-17A or IL-5. The TLR4 ligand INI-2002 drove a greater development of Th17 cells along with somewhat lower levels of Th1 and Th2, resulting in a mixed T helper cell response. The combination of the two ligands resulted in a response very similar to that of INI-2002 alone, suggesting that INI-2002 dominates the adjuvant activity, at least in the absence of alum.

The alum-containing formulations in the absence of TLR ligands (green symbols) failed to elicit IFNγ- or IL-17A-producing T cells, resulting in a Th2-dominated immune response (evidenced by IL-5 production and no detectable IFNγ or IL-17A), as expected. The addition of TLR7/8 ligand INI-4001, either alone (blue symbols) or in combination with INI-2002 (purple symbols), resulted in enhanced development of IFNγ-producing Th1 cells above what was seen in the absence of alum. Again, the TLR4 ligand INI-2002 promoted the additional development of IL-17A-producing (Th17) cells, even in the presence of alum. A moderate level of IL-5 was also produced by most of the groups, resulting in a mixed T cell phenotype that corroborates the mix of antibody isotypes that were previously noted. The exception to this occurred when both TLR ligands were added to alum, which abrogated nearly all IL-5-producing cells while promoting the development of Th1 and Th17 cells instead.

To summarize these results, the addition of TLR ligands to alum results in an additive mix of what is seen from each independently. Th2 cells develop due to alum, Th1 cells from INI-4001, and both Th1 and Th17 cells from INI-2002. However, in the case where both TLR ligands are used with alum, they effectively eliminate the alum-induced Th2 cell development, perhaps due to increased development of Th1 and Th17 cells. When alum was not included in the vaccines (triangles), the overall level of antigen-specific T cell cytokines was lower, suggesting that alum enhances effector T cell development without notably skewing the response. The enhancement of antibody titers (most noticeable in formulations with INI-4001) along with the enhancement of Th1 and Th17 CD4^+^ T cell responses suggests that a combination of adjuvants will offer enhanced protection against viral infection over what each of these adjuvants would promote on their own.

### 3.9. Improved Neutralizing Antibody Potential with the Combination of Alum and TLR Ligands

An important correlate of protection in COVID-19 infection is the presence of neutralizing antibody titers, and in order to enter a cell, the spike protein must bind to the receptor ACE2. Therefore, the serum collected 28 days after booster vaccination was tested for the ability to prevent the binding of ACE2 to the spike RBD, as well as the full spike glycoprotein (trimer), as an indication of virus-neutralizing potential (Figure 10). In general, the neutralization potential correlated with the level of RBD-specific IgG antibody titers (compared with Figure 7), with a few notable exceptions. Serum antibodies from mice vaccinated with AP and both TLR agonists (purple squares) had notably lower neutralizing potential than would be expected based on the anti-RBD antibody titers. Also striking is the low neutralization potential of serum antibodies after vaccination with INI-2002 or the combination of INI-2002 and INI-4001 in the absence of alum (triangles). While these vaccines resulted in high anti-RBD antibody titers, these sera were less effective at preventing the binding of RBD or Spike to hACE2 than would be expected from those titers (compare Figure 7B and Figure 10). Therefore, it is notable that the presence of alum greatly improved the quality of the antigen-specific antibodies, as judged by their ability to block the spike protein.

## 4. Conclusions

We developed novel adjuvanted SARS-CoV-2 subunit vaccines by co-delivering novel synthetic TLR7/8 and/or TLR4 ligands and RBD by adsorption to aluminum salts. Although RBD did not adsorb to alhydrogel (AH) on its own, an increase in pH and the addition of the TLR ligands facilitated its adsorption, and AH and AP proved to be similarly effective for eliciting an immune response. In vivo, adding TLR ligands not only enhanced neutralizing antibody titers but also enhanced and modulated the T cell response, shifting it from a primarily Th2-mediated response towards a more Th1 (INI-4001), or a mix of Th17 and Th1 (INI-2002). The change in CD4^+^ T cell phenotype is likely to be advantageous for fighting certain pathogens more effectively. For example, a Th1 response is preferred for the fight against many viral pathogens, and in some cases, a Th2-dominated response may prove ineffective or even dangerous [45]. The ability to skew the T cell response through the combination of adjuvants allows a vaccine strategy to tailor protection to the specific pathogen better. In the case of SARS-CoV-2, the correlates of protection are still being fully defined, but too strong of a Th2 response in the absence of Th1 cells may promote harmful lung inflammation in the situation where neutralizing antibody levels have declined below a level of disease prevention.

A broadened antibody repertoire and a strong cell-mediated response will be important to protect against variants that have mutated to escape recognition. We see evidence that the mix of either alum with one or both TLR ligands promotes both of these. The increased concentration of cytokines measured in Figure 9 is likely representative of a larger number of effector T cells, while improved antibody titers will offer broader protection (Figure 7 and Figure 10). The combination adjuvants allow for these robust responses from the RBD antigen, which shows serum antibody titers equivalent to unvaccinated animals in the absence of adjuvants. The relatively easy production and purification of the RBD antigen allows for less expensive and more rapid production of new antigens from emerging mutant strains of the virus compared with the full spike trimer antigen. Combined with storage requirements no lower than 4 °C, using the spike RBD antigen with alum and synthetic TLR agonists represents a distinct advantage in a pandemic setting, especially in areas where cost and storage conditions can prove to be a significant barrier.

## Figures and Tables

**Figure 1 vaccines-12-00021-f001:**
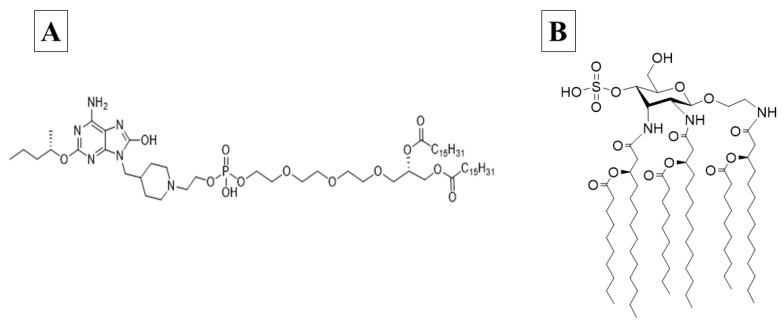
Structures of INI-4001 (**A**), a lipidated oxoadenine and a TLR 7/8 agonist, and INI-2002 (**B**), a synthetic TLR 4 agonist.

**Figure 2 vaccines-12-00021-f002:**
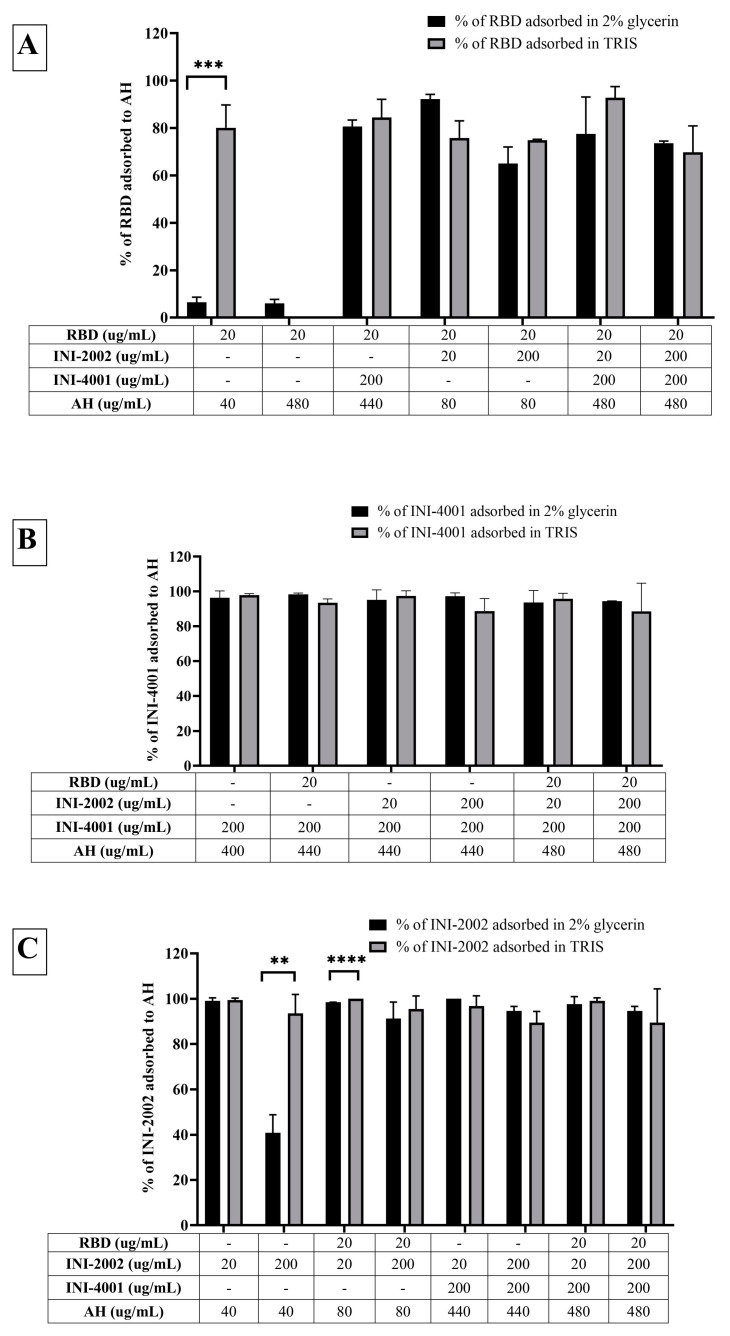
Adsorption efficiencies of (**A**) RBD, (**B**) INI-4001, and (**C**) INI-2002 to AH. RDB did not adsorb to AH in 2% glycerin. The usage of TRIS buffer at pH 8.1 significantly improved the adsorption efficiency of RBD to AH. INI-4001 and INI-2002 adsorbed to AH in both 2% glycerin and TRIS buffer. A one-way ANOVA with a post hoc Tukey’s test was used to compare between groups. Asterisks indicate statistical differences between relevant groups; ** = *p* < 0.01, *** = *p* < 0.001, **** = *p* < 0.0001.

**Figure 3 vaccines-12-00021-f003:**
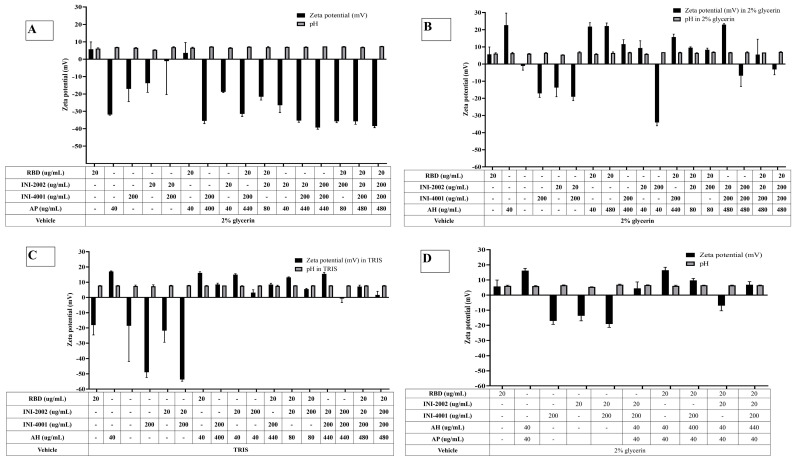
Zeta potential and pH of mixtures of RBD, INI-4001, and INI-2002 with (**A**) AP, (**B**) AH in 2% glycerin as formulation vehicle, (**C**) AH in TRIS buffer as the formulation vehicle, and (**D**) Combination of AH and AP.

**Figure 4 vaccines-12-00021-f004:**
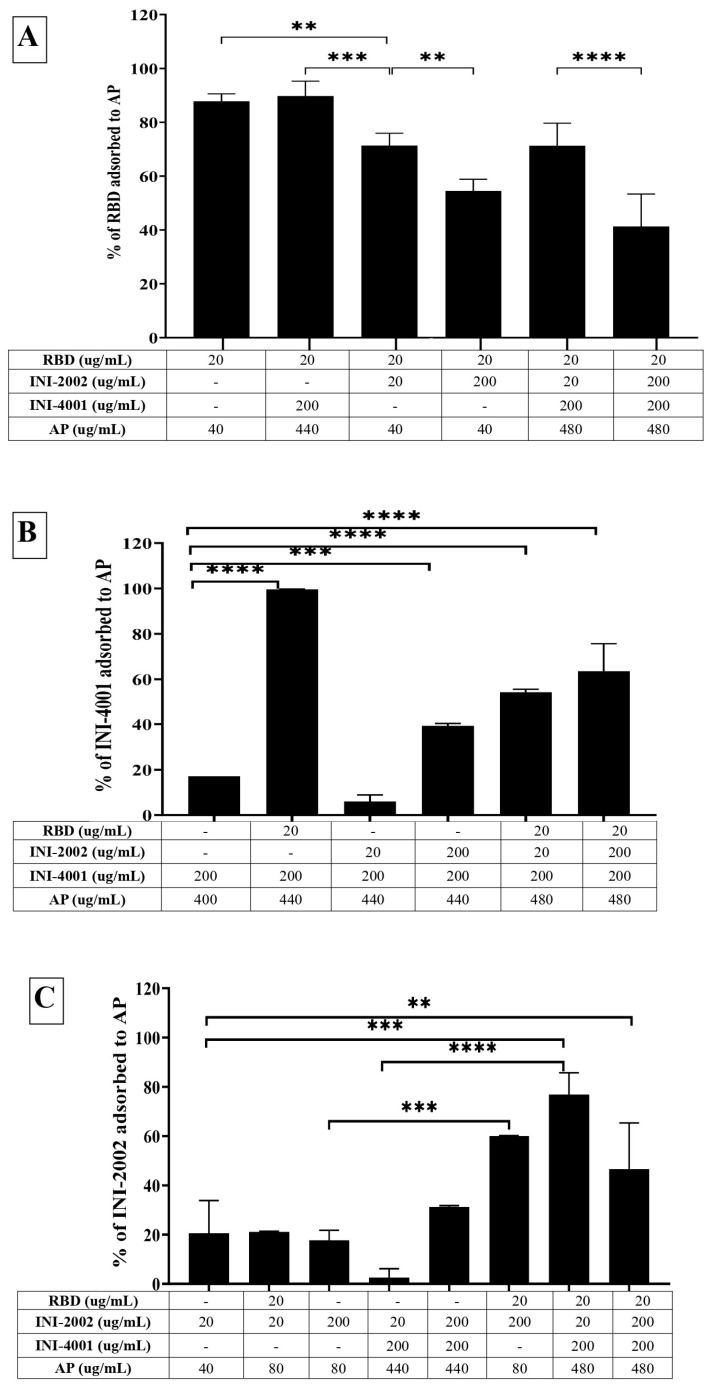
Adsorption efficiencies of (**A**) RBD, (**B**) INI-4001, and (**C**) INI-2002 to AP. The results suggest that RBD adsorbed to AP and facilitated the adsorption of INI-4001 and INI-2002. A one-way ANOVA with a post hoc Tukey’s test was used to compare between groups. Asterisks indicate statistical differences between relevant groups; ** = *p* < 0.01, *** = *p* < 0.001, **** = *p* < 0.0001.

**Figure 5 vaccines-12-00021-f005:**
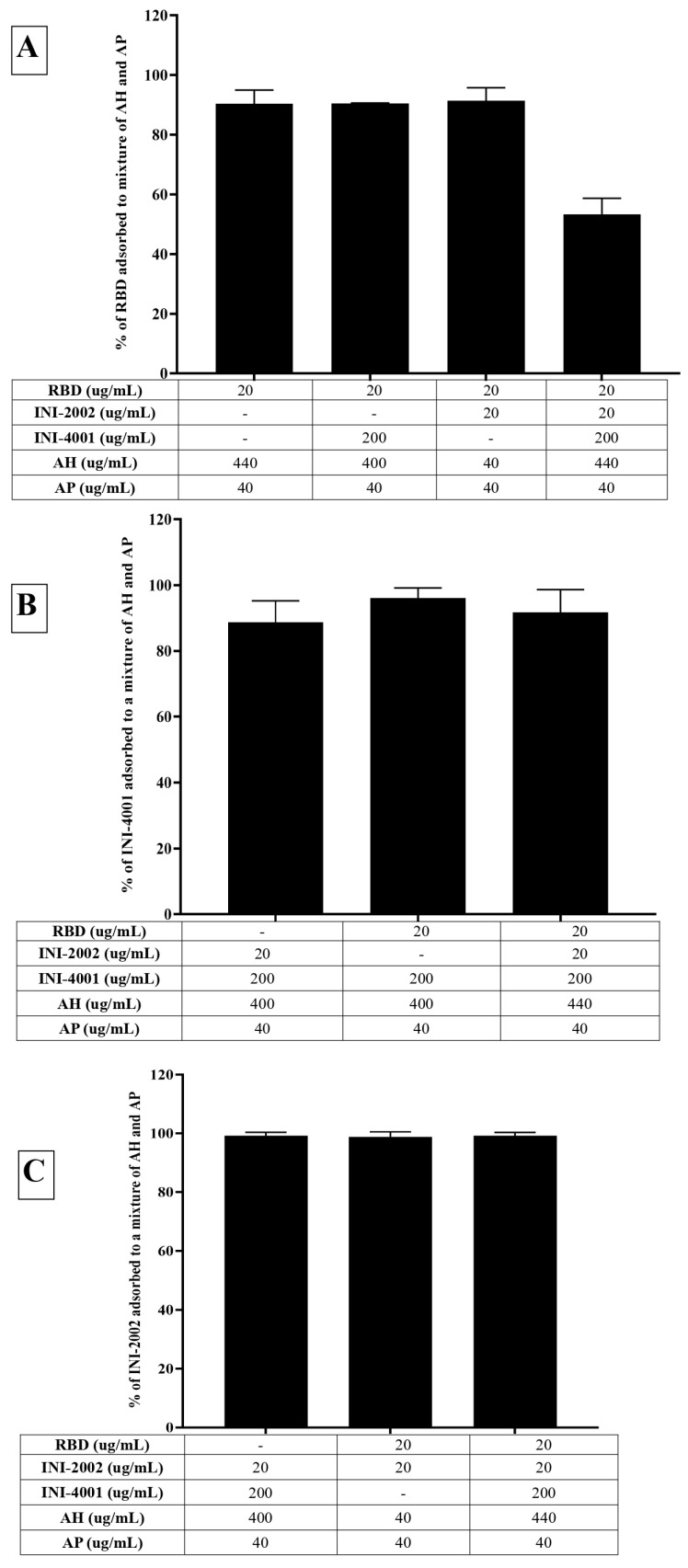
Adsorption efficiencies of (**A**) RBD, (**B**) INI-4001, and (**C**) INI-2002 to a mixture of AH and AP. RBD, INI-4001, and INI-2002 adsorbed to a mixture of AH and AP. A one-way ANOVA with a post hoc Tukey’s test was used to compare between groups. Asterisks indicate statistical differences between relevant groups.

**Figure 6 vaccines-12-00021-f006:**
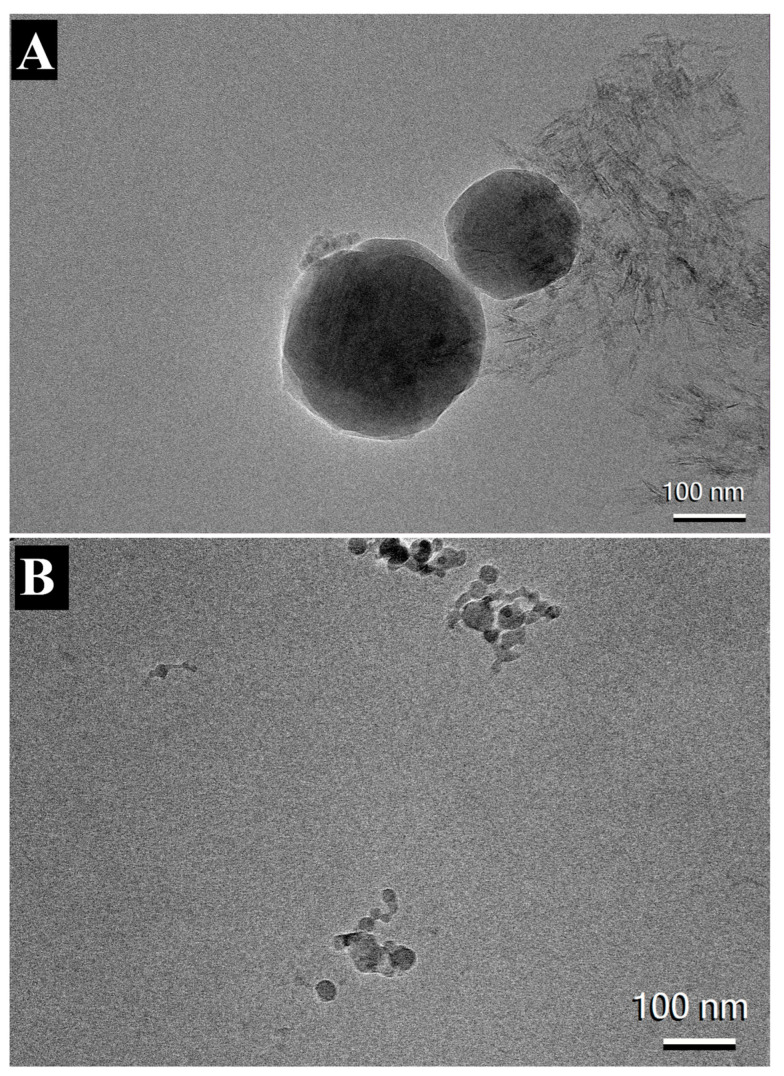
CryoTEM image of AH (**A**) and AP (**B**) adsorbed with RBD, INI-4001, and INI-2002.

**Figure 7 vaccines-12-00021-f007:**
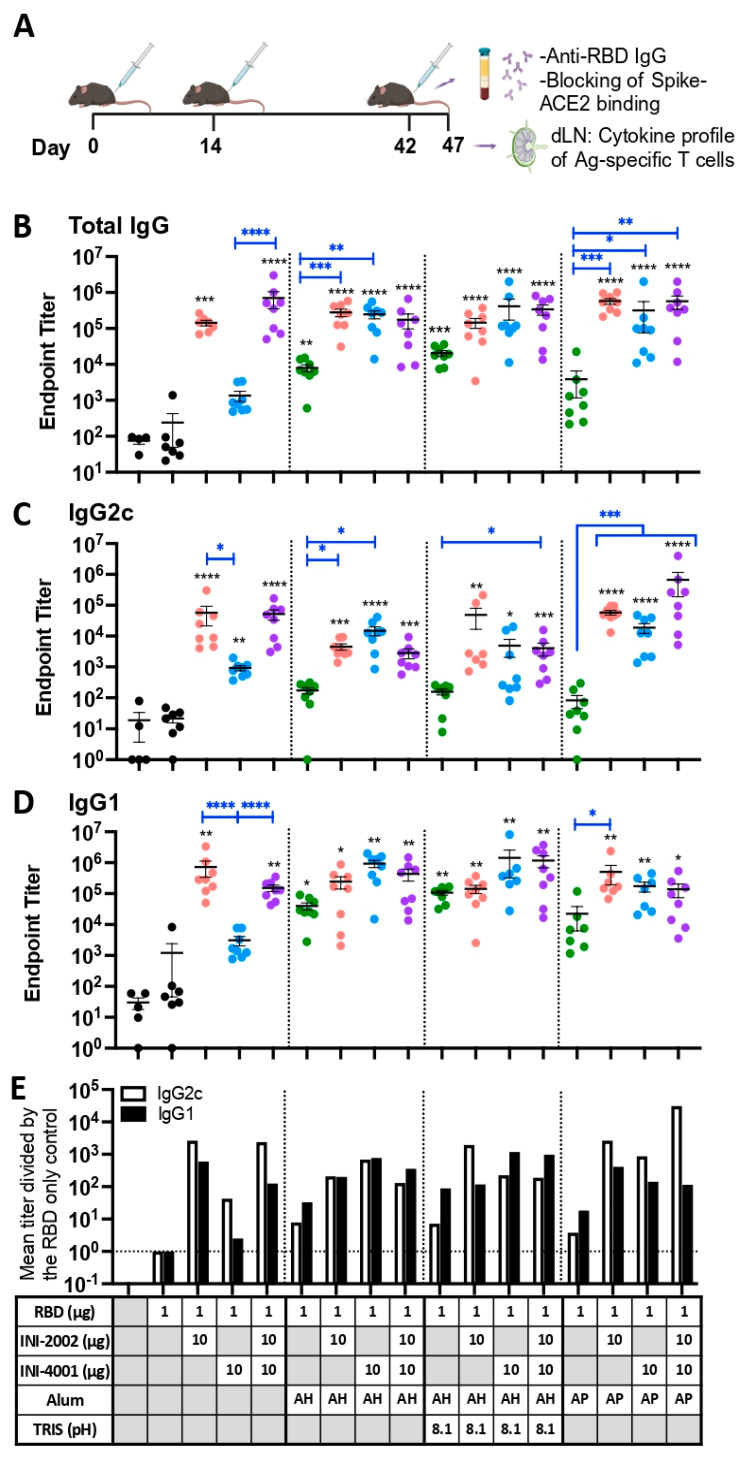
**RBD-specific serum IgG antibody titers.** (**A**) Schematic of the experimental timeline (created with BioRender.com). (**B**–**D**) Titer of RBD-binding antibodies in serum from 28 days after 2nd vaccination. (**B**) IgG (**C**) IgG2c isotype (**D**) IgG1 isotype. (**E**) relative fold change in the mean titer of RBD-specific IgG2c (open) or IgG1 (filled) over the mean titer of the RBD alone group. Black lines indicate the mean and standard error of the mean for each group, while individual mice are represented by symbols. Black asterisks designate significant differences (by one-way ANOVA) compared with the RBD antigen-only group (solid black circles). Symbols for formulations containing only alum are green, those with INI-2002 are red, INI-4001 blue, and both together are purple. Blue lines and asterisks specify significant differences among groups receiving the same type of formulation. * = *p* < 0.05, ** = *p* < 0.01, *** = *p* < 0.001, **** = *p* < 0.0001.

**Figure 8 vaccines-12-00021-f008:**
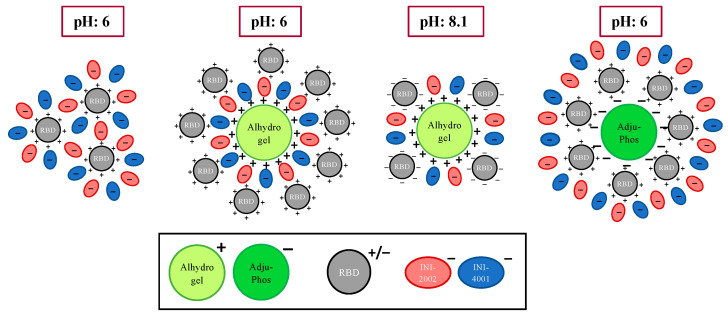
Schematic representation of hypothetical interactions of RBD, INI-4001, and INI-2002 with AH or AP based upon charge.

**Figure 9 vaccines-12-00021-f009:**
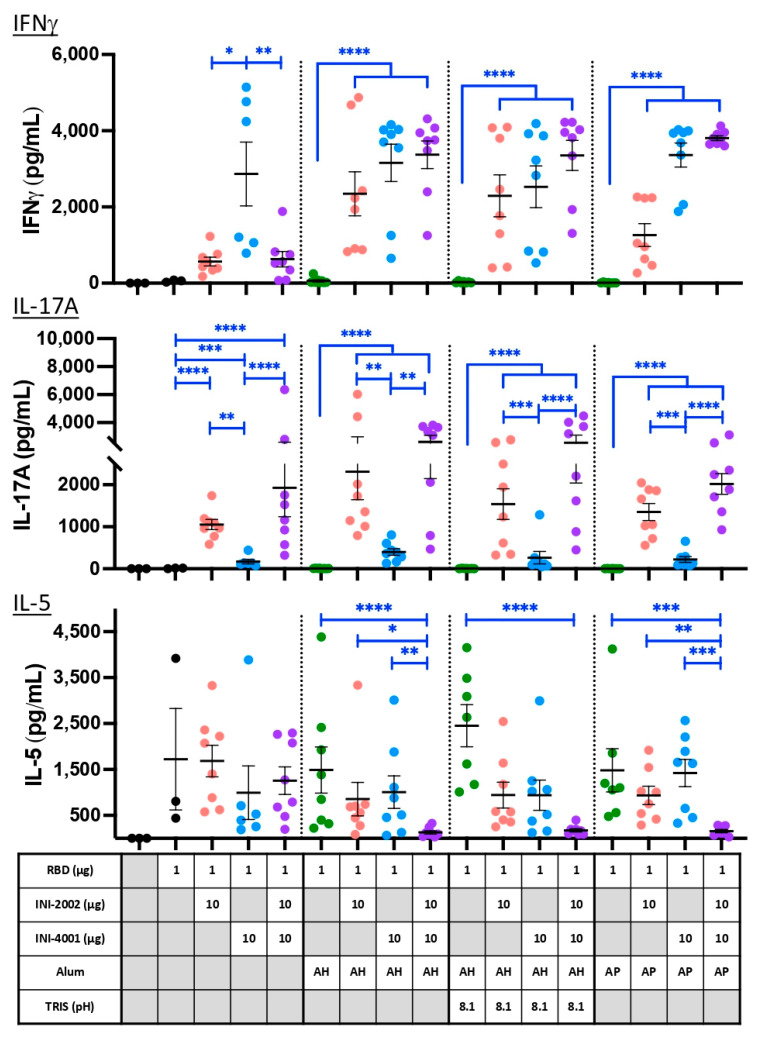
**Cytokine release from RBD-specific T cells upon restimulation.** Cytokine concentrations in supernatants recovered from a 72 h culture of ex vivo dLN cell suspensions stimulated with 10 µg/mL RBD protein. Five days after a third vaccination, cells were recovered and cultured to assess the phenotype of RBD-specific T cells. Black lines indicate the mean and standard error of the mean for each group, while individual mice are represented by symbols. Symbols for formulations containing only alum are green, those with INI-2002 are red, INI-4001 blue, and both together are purple. Black asterisks designate significant differences compared with the RBD antigen-only group (black). Blue lines and asterisks specify significant differences among groups receiving the same type of formulation (aqueous or alum type) * = *p* < 0.05, ** = *p* < 0.01, *** = *p* < 0.001, **** = *p* < 0.0001.

**Figure 10 vaccines-12-00021-f010:**
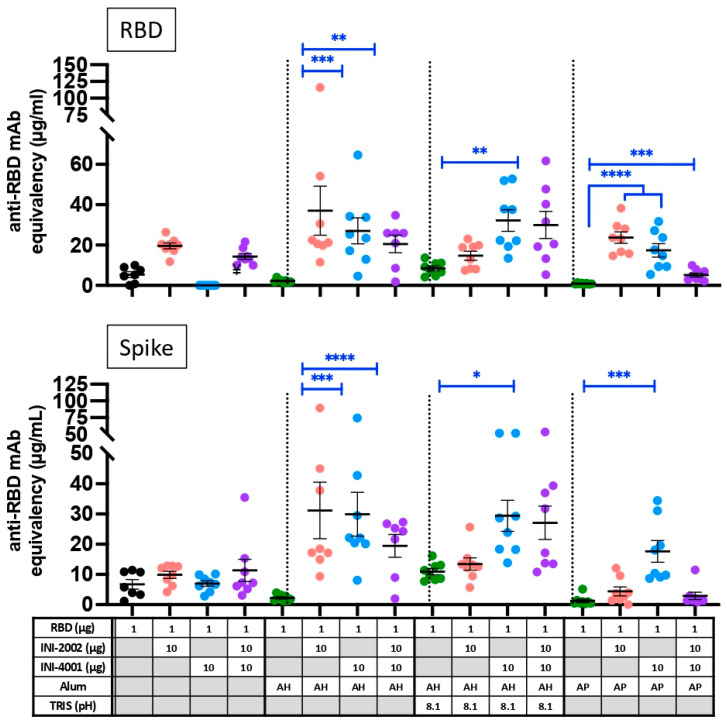
**Surrogate neutralization assay with serum antibodies from 14 days after booster vaccination.** The ability to block the interaction between human ACE2 and RBD (**top**) or trimerized spike ectodomain (**bottom**) was assessed using a competitive ELISA-based assay. Black lines indicate the mean of the group, and symbols represent data from individual animals. Symbols for formulations containing only alum are green, those with INI-2002 are red, INI-4001 blue, and both together are purple. Blue lines and asterisks indicate significant differences among groups receiving the same type of formulation (aqueous or alum type). ¥ indicates samples that were below detection in this assay. * = *p* < 0.05, ** = *p* < 0.01, *** = *p* < 0.001, **** = *p* < 0.0001.

**Table 1 vaccines-12-00021-t001:** Formulation composition of the various SARS-CoV-2 sub-unit vaccines containing RBD and TLR ligands.

Formulation Composition	Concentration of the Adsorbant (µg/mL)		Concentration of the Adsorbate (µg/mL)					
	AH	AP	RBD		INI-4001		INI-2002	
			2% Glycerin	TRIS Buffer	2% Glycerin	TRIS Buffer	2% Glycerin	TRIS Buffer
RBD	0	0	20	20	0	0	0	0
2% glycerin	0	0	0	0	0	0	0	0
INI-4001 only	0	0	0	0	200.00	200.00	0	0
INI-2002 only	0	0	0	0	0	0	20.00	20.00
INI-4001 + INI-2002	0	0	0	0	200.00	200.00	20.00	20.00
Alhydrogel	480	0	0	0	0	0	0	0
RBD + Alhydrogel (1:2)	40	0	20	20	0.00	0.00	0	0
RBD + Alhydrogel (1:24)	480	0	20	20	0.00	0.00	0	0
INI-4001 + Alhydrogel	400	0	0	0	200.00	200.00	0	0
INI-2002 + Alhydrogel	40	0	0	0	0.00	0.00	20.00	20.00
INI-2002 + Alhydrogel (10 times)	40	0	0	0	0.00	0.00	200.00	200.00
INI-4001 + INI-2002 + Alhydrogel	440	0	0	0	200.00	200.00	20.00	20.00
INI-4001 + INI-2002 + Alhydrogel (10:10)	440	0	0	0	200.00	200.00	200.00	200.00
RBD + INI-4001 + Alhydrogel	440	0	20	20	200.00	200.00	0	0
RBD + INI-2002 + Alhydrogel	80	0	20	20	0.00	0.00	20.00	20.00
INI-2002 (10X) + Alhydrogel	80	0	20	20	0.00	0.00	200	200
RBD + INI-4001 + INI-2002 + Alhydrogel	480	0	20	20	200.00	200.00	20.00	20.00
RBD + INI-4001 + INI-2002 (10X) + Alhydrogel	480	0	20	20	200.00	200.00	200.00	200.00
Adju-Phos	0	40	0	0	0	0	0	0
RBD + Adju-Phos	0	40	20	20	0.00	0.00	0	0
INI-4001 + Adju-Phos	0	400	0	0	200.00	0.00	0	0
INI-2002 + Adju-Phos	0	40	0	0	0.00	0.00	20.00	0.00
INI-2002 (10X) + Adju-Phos	0	40	0	0	0.00	0.00	200.00	200.00
INI-4001 + INI-2002 + Adju-Phos	0	440	0	0	200.00	0.00	20	0
INI-4001 + INI-2002 + Adju-Phos (10:10)	0	440	0	0	200.00	0.00	200	200
RBD + INI-4001 + Adju-Phos	0	440	20	20	200.00	0.00	0	0
RBD + INI-2002 + Adju-Phos	0	80	20	20	0.00	0.00	20.00	0.00
RBD + INI-2002 (10X) + Adju-Phos	0	80	20	20	0.00	0.00	200	0
RBD + INI-4001 + INI-2002 + Adju-Phos	0	480	20	20	200.00	0.00	20	0
RBD + INI-4001 + INI-2002 (10X) + Adju-Phos	0	480	20	20	200.00	0.00	200	0
Alhydrogel + Adju-Phos	440	40	0	0	0	0	0	0
RBD + Alhydrogel + Adju-Phos	440	40	20	20	0.00	0.00	0	0
INI-4001 + INI-2002 + Alhydogel + Adju-Phos	440	40	0	0	200.00	0.00	20.00	0.00
RBD + INI-4001 + Alhydrogel + Adju-Phos	400	40	20	20	200.00	0.00	0	0
RBD + INI-2002 + Alhydrogel +Adju-Phos	40	40	20	20	0.00	0.00	20.00	0.00
RBD + INI-4001+ INI-2002 + Alhydrogel + Adju-Phos	440	40	20	20	200.00	0.00	20.00	0.00

## Data Availability

The data presented in this study are available from the corresponding author upon request.

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
