# Peer review of "Co-Delivery of Novel Synthetic TLR4 and TLR7/8 Ligands Adsorbed to Aluminum Salts Promotes Th1-Mediated Immunity against Poorly Immunogenic SARS-CoV-2 RBD"

_vaccines, 2023, doi:10.3390/vaccines12010021_

Round 1

Reviewer 1 Report

Comments and Suggestions for Authors

Fig 3 does not have good quality and needs to be prepared in better way . In table 1 row 17 RBD was mentioned 2 times

Author Response

  1. Fig 3 does not have good quality and needs to be prepared in better way. In table 1 row 17 RBD was mentioned 2 times.

Response: Thank you for the careful review of the manuscript. We have replaced Fig 3 in the manuscript with a higher resolution image (300 dpi). We deleted the extra “RBD” accordingly in table 1 row 17.

Reviewer 2 Report

Comments and Suggestions for Authors

Siram et al describe the manufacturing of novel adjuvanted SARS-CoV-2 subunit vaccines by co-delivering novel synthetic TLR7/8 and /or TLR4 ligands and RBD protein adsorbed to aluminum salts, alhydrogel and /or adju-phos. The authors produced and tested different combinations of adjuvants and aluminum salts, which were extensively characterized by means of physicochemical characteristics. Eventually, authors showed that those novel vaccines were able to induce cellular as well as humoral immune responses after immunization of C57BL/6 mice. 

Major concerns:

1. Since the suggested vaccines are new and they have a huge amount of adjuvants, authors should have tested their biocompatibilty by measuring biochemical markers in serum as well as induction of inflammation (systemic or at the site of injection).

2. The success of a proposed vaccine is dependent on its capability to induce the differentation of antigen-specific T and B cell memory populations. Such experiments are missing.

3. Authors in lines 261-262 based on bibliography support that "alum promotes recruitment of dendritic cells and macrophages to the draining lymph nodes..". However, such experiment showing APCs infiltration into dLNs is missing based on the fact that the proposed vaccines are not just alum but they contain also TLR ligands.

4. Results section, Line 269-270: Authors should provide data regarding adsorption in different weight ratios tested in a Supplementary table, in order to be clear why the ratio 1:2 was selected.

5. In line 273, what authors mean by writing ".. to ensure adsorption equilibrium has been reached."? 

6. Lines 278-279. This sentence needs the respective reference(s).

7. Provide an immunization scheme for reasons of clarity, since many different formulations were used.

8. Lines 413-414: How the uptake of particles by APCs was conducted? It is known surface charge in cell membrane plays a role as well as particle's size. Please discuss.

9. Paragraph 3.8: Authors should provide the numerical values of the produced cytokines inside text along with significance for reasons of clarity.

10. Lines 429-430: Authors describe that they used cells obtained from spleen and LNs but the results the presented are obtained from LN cultures only. Please explain. The same observation is also for the respective paragraph in Materials and Methods section.

11. Conclusion: The authors did not conclude in a specific formulation and explain why, but they give a very general conclusion. Please specify based on the results given. Are AH or AP both efficient to induce immune response? Which are the drawbacks and the benefits of the formulations?

12. All figure legends should be more descriptive. Also, figure 3 should be larger. 

Minor commments:

1. Line 126: Provide information regarding AH and AP (Cat. No, Company etc.)

2. Line 133: How did authors estimate that the particle size was <200 nm?

3. Line 182: Provide information regarding Coomassie plus kit.

4. Reference 29: Provide all the information. 

Author Response

Comments and Suggestions for Authors

Siram et al describe the manufacturing of novel adjuvanted SARS-CoV-2 subunit vaccines by co-delivering novel synthetic TLR7/8 and /or TLR4 ligands and RBD protein adsorbed to aluminum salts, alhydrogel and /or adju-phos. The authors produced and tested different combinations of adjuvants and aluminum salts, which were extensively characterized by means of physicochemical characteristics. Eventually, authors showed that those novel vaccines were able to induce cellular as well as humoral immune responses after immunization of C57BL/6 mice. 

Major concerns:

  1. Since the suggested vaccines are new and they have a huge amount of adjuvants, authors should have tested their biocompatibilty by measuring biochemical markers in serum as well as induction of inflammation (systemic or at the site of injection).

Response: Thank you for your comment and suggestion. We agree that the dose and type of adjuvants can impact inflammation and reactogenicity (local and systemic). We selected the individual and combination dose of the TLR adjuvants based on historical optimized dosing studies with other antigens across various formulations. Similar mg/kg dosing has been used in mice, rats, pigs and NHPs with these adjuvants, with no signs of reactogenicity or toxicity. We recently completed an IND-enabling GLP toxicology study with INI-4001/alum administered IM in rats and pigs with no dose limiting toxicity noted. In the course of the studies reported herein, no sign of inflammation or ill effects was observed in the animals. We have added wording to the section 2.7 of the materials and methods (lines 202-204) referring to the fact we have tested these without signs of inflammation or reactogenicity for readers with this same concern. 

Evaluating local biomarkers of adjuvant impacts on innate immunity a the injection site and draining lymph nodes is an important next step in the further development and understanding of vaccine MOA, but is outside the scope of the current manuscript (focused on formulation characterization and in vivo immunogenicity).

  1. The success of a proposed vaccine is dependent on its capability to induce the differentiation of antigen-specific T and B cell memory populations. Such experiments are missing.

Response 1: We thank the reviewer for this excellent suggestion. The current studies show that the combination TLR agonist/alum vaccines cause the development of a strong effector B and T cell response (RBD-specific antibody titers and antigen-specific induction of T cell-associated cytokines, Figs 7, 9), which anticipate to result in memory cells. However, direct measurement of memory T and B cell populations was not included in these studies. This manuscript serves to first introduce these vaccine formulations, and centers on their development and characterization. We feel that the comparison of memory development from these novel vaccine formulations in comparison to benchmarks (AS04 for example) would be a valuable addition to our planned future publication centered on adjuvant combinations MOA.

  1. Authors in lines 261-262 based on bibliography support that "alum promotes recruitment of dendritic cells and macrophages to the draining lymph nodes..". However, such experiment showing APCs infiltration into dLNs is missing based on the fact that the proposed vaccines are not just alum but they contain also TLR ligands.

Response: In this sentence, we referred to alum’s reported MOA to support our rationale for testing the alum-containing formulations. However, the placement of this statement at the beginning of the results section implied that we would test the ability of the alum/TLR formulations to promote recruitment in our studies. Therefore, this statement seems more appropriate in the context of the discussion of antibody titer results, and we’ve moved it to later in the manuscript (see line 447-450), replacing this statement at the beginning of the results and discussion section accordingly (see lines 269-274 of revised manuscript). We appreciate that the reviewer brought up this point, and we do feel that the comparison of APC recruitment by alum with and without the TLR ligands will be an excellent addition to future investigations into the mechanism by which the combination of alum with TLR ligands alters the immune response.

  1. Results section, Line 269-270: Authors should provide data regarding adsorption in different weight ratios tested in a Supplementary table, in order to be clear why the ratio 1:2 was selected.

Response: Thank you for this suggestion. We added this information as Figure S1.

  1. In line 273, what authors mean by writing ".. to ensure adsorption equilibrium has been reached."? 

Response: We revised the line in the manuscript in hopes of making our meaning clearer. Please find the new line below and at lines 281-282 of the revised manuscript.

“Although complete adsorption was achieved in 30 minutes, the duration of mixing was increased to 60 minutes to ensure complete adsorption.”.

  1. Lines 278-279. This sentence needs the respective reference(s).

Response: Thank you. This reference has been added to the revised version of the manuscript (line 288).

  1. Provide an immunization scheme for reasons of clarity, since many different formulations were used.

Response: We have added a schematic that describes the vaccination experimental timeline to Figure 7, as part A. Hopefully this will make the experiment easier for readers to follow.

  1. Lines 413-414: How the uptake of particles by APCs was conducted? It is known surface charge in cell membrane plays a role as well as particle's size. Please discuss.

Response: We thank the reviewer for bringing up this very interesting point.  We agree that both charge and size can play a very important role in uptake by APC. Alhydrogel and adju-phos are usually positively and negatively charged, respectively, in the pH range we use for vaccination. While we have tracked the change in overall particle charge following the adsorption of TLR agonist(s) to alhydrogel (Figure 3), we can only speculate what the surface charge is.

We have added to our Results and Discussion section a more extensive discussion of the impact of adding the TLR ligands to each type of alum, and of the hypothetical particle compositions shown in Figure 8. It is difficult to tease apart the impacts of particle size and surface charge, combined with the likely increase in recruitment of APCs to the injection site due to the effects of alum itself. We hope these alterations and additions (lines 438-458) serve to engage readers in a more insightful interpretation of the data.

  1. Paragraph 3.8: Authors should provide the numerical values of the produced cytokines inside text along with significance for reasons of clarity.

Response: We have added Table S1 to the supplement that reports these data. We felt adding them directly to the text would compromise its readability due to the large amount of data, and that the graphs in Figure 9 allow for an adequate estimation of the numbers to follow with the text. We hope the reviewer will agree.

  1. Lines 429-430: Authors describe that they used cells obtained from spleen and LNs but the results the presented are obtained from LN cultures only. Please explain. The same observation is also for the respective paragraph in Materials and Methods section.

Response: Thank you for noticing this discrepancy! We only included the dLN cells in this experiment because we have consistently found they produce a higher antigen-specific response than splenocytes. We have deleted the references to spleen cells (which we have often included in other, similar experiments done in our laboratory, sorry for the mistake and confusion). See lines 211-212 and 466.

  1. Conclusion: The authors did not conclude in a specific formulation and explain why, but they give a very general conclusion. Please specify based on the results given. Are AH or AP both efficient to induce immune response? Which are the drawbacks and the benefits of the formulations?

Response: Thank you for pointing out that the conclusions section could be improved upon. We have expanded it to more distinctly discuss the advantage of using a combination of alum and TLR ligands, and specifically mention that both types of alum performed similarly in combination with TLR agonists.

  1. All figure legends should be more descriptive. Also, figure 3 should be larger. 

Response: In the revised manuscript, we added additional information to the legends. We have also replaced Fig 3 in the manuscript with a higher-resolution image (300 dpi) for publication.

Minor commments:

  1. Line 126: Provide information regarding AH and AP (Cat. No, Company etc.)

Response: The catalog numbers and company name for AH and AP have been added to the revised manuscript (lines 131-132).

  1. Line 133: How did authors estimate that the particle size was <200 nm?

Response: The particle size was measured using Malvern Zetasizer Nano-ZS (Malvern Panalytical, Malvern, UK). The same information been added to the revised manuscript (lines 138-139).

  1. Line 182: Provide information regarding Coomassie plus kit.

Response: The information regarding the Coomassie plus kit has been added to the revised manuscript (lines 188-189).

  1. Reference 29: Provide all the information. 

Response: Thank you for finding this omission, we’ve included the complete reference in the revised manuscript.

Reviewer 3 Report

Comments and Suggestions for Authors

o-delivery of novel synthetic TLR4 and TLR7/8 ligands adsorbed to aluminum salts promotes Th1-mediated immunity against poorly immunogenic SARS-CoV-2 RBD

vaccines-2752745

The authors have developed an adjuvanted subunit vaccine against SARS-CoV-2 using recombinant receptor-binding domain (RBD) of spike with synthetic adjuvants targeting TLR7/8 (INI-4001) and TLR4 (INI-2002), co-delivered with aluminum hydroxide (AH) or aluminum phosphate (AP).  In vivo vaccinations of AP and AH  salts of the RBD in C57BL/6 mice, both aluminum salts promoted Th2-mediated immunity when used as the sole adjuvant. Interestingly, the co-delivery with TLR4 and/or TLR7/8 ligands efficiently promoted a switch to Th1-mediated immunity. The promising in vivo results can easily provide useful evidence for proving that the addition of a TLR7/8 and/or TLR4 agonist to a subunit vaccine containing RBD antigen and alum is a promising strategy for driving a Th1 response and neutralizing antibody titers targeting SARS-CoV-2.

The article is a very interesting read and important information and validity on how TLR7/8 and/or TLR4 agonist to a subunit vaccine are presented by the authors. The study is well-designed and backed up by animal results. The physiochemical analysis of the formulation is been conducted in a very systematic and thorough way. 

Minor comments:

1. Please improve the figure presentations. Please use software like Prism for the bar graphs, or any similar software. 

2. The font in the figures should match the template of the journal.

3. Similarly the font in Table 1 should match the template of the journal. 

Comments on the Quality of English Language

Minor editing of English language required

Author Response

Co-delivery of novel synthetic TLR4 and TLR7/8 ligands adsorbed to aluminum salts promotes Th1-mediated immunity against poorly immunogenic SARS-CoV-2 RBD

vaccines-2752745

The authors have developed an adjuvanted subunit vaccine against SARS-CoV-2 using recombinant receptor-binding domain (RBD) of spike with synthetic adjuvants targeting TLR7/8 (INI-4001) and TLR4 (INI-2002), co-delivered with aluminum hydroxide (AH) or aluminum phosphate (AP).  In vivo vaccinations of AP and AH  salts of the RBD in C57BL/6 mice, both aluminum salts promoted Th2-mediated immunity when used as the sole adjuvant. Interestingly, the co-delivery with TLR4 and/or TLR7/8 ligands efficiently promoted a switch to Th1-mediated immunity. The promising in vivo results can easily provide useful evidence for proving that the addition of a TLR7/8 and/or TLR4 agonist to a subunit vaccine containing RBD antigen and alum is a promising strategy for driving a Th1 response and neutralizing antibody titers targeting SARS-CoV-2.

The article is a very interesting read and important information and validity on how TLR7/8 and/or TLR4 agonist to a subunit vaccine are presented by the authors. The study is well-designed and backed up by animal results. The physiochemical analysis of the formulation is been conducted in a very systematic and thorough way. 

Minor comments:

  1. Please improve the figure presentations. Please use software like Prism for the bar graphs, or any similar software. 

Response: Thank you for your comment. The revised manuscript has been updated with better-quality figures that are more suitable for publication.

  1. The font in the figures should match the template of the journal.

Response: In the revised manuscript, the font in the figures matches the template of MDPI.

  1. Similarly the font in Table 1 should match the template of the journal. 

Response: In the revised manuscript, the font of Table 1 matches the template of MDPI.

Reviewer 4 Report

Comments and Suggestions for Authors

This study presents an intriguing exploration into developing an enhanced adjuvant system, co-delivering INI-4001 and INI-2002 with AH and AP, aimed at augmenting the SARS-CoV-2 vaccine's efficacy. The authors have demonstrated that incorporating a TLR7/8 and/or TLR4 agonist into a subunit vaccine containing RBD antigen and alum could be a promising approach to stimulate a Th1 response and elevate neutralizing antibody titers against SARS-CoV-2. The implications of this work for vaccine development are potentially significant. However, for a more comprehensive evaluation suitable for publication, certain critical aspects need addressing:

  1. Adsorption Study Stability: In Figure 2, the adsorption study is informative, but it leaves a gap regarding the long-term stability of the vaccine or adjuvant system. Could the authors clarify whether they assessed this aspect?

  2. Statistical Analysis in Figures: Many figures, including crucial data representations, lack significance analysis (such as P-values). This omission makes it challenging to draw definitive conclusions from the results. Including statistical analysis would greatly enhance the validity and clarity of the findings.

  3. TEM Image Clarity: The TEM images in Figure 6B do not clearly show the presence of AP. Could the authors provide more detail or clarification on this aspect?

  4. Neutralization Capability of Antibodies: While the study effectively measures antibody titers induced by the vaccines, it does not address the critical factor of how effectively these antibodies neutralize the virus. Evaluating the neutralizing capability of the antibodies would significantly strengthen the study's conclusions and relevance.

Author Response

This study presents an intriguing exploration into developing an enhanced adjuvant system, co-delivering INI-4001 and INI-2002 with AH and AP, aimed at augmenting the SARS-CoV-2 vaccine's efficacy. The authors have demonstrated that incorporating a TLR7/8 and/or TLR4 agonist into a subunit vaccine containing RBD antigen and alum could be a promising approach to stimulate a Th1 response and elevate neutralizing antibody titers against SARS-CoV-2. The implications of this work for vaccine development are potentially significant. However, for a more comprehensive evaluation suitable for publication, certain critical aspects need addressing:

  1. Adsorption Study Stability: In Figure 2, the adsorption study is informative, but it leaves a gap regarding the long-term stability of the vaccine or adjuvant system. Could the authors clarify whether they assessed this aspect?

Response: We agree that the long-term stability of the vaccine or adjuvant system is important, especially after adsorption to the aluminum salts. Although, the individual aqueous formulations of the adjuvants, INI-4001 and INI-2002, are stable for >1 year at 4°C, we have not evaluated long-term and accelerated stability of the alum adsorbed vaccine formulations since further analytical assay development is necessary to develop a stability indicating assay with this formulation. 

  1. Statistical Analysis in Figures: Many figures, including crucial data representations, lack significance analysis (such as P-values). This omission makes it challenging to draw definitive conclusions from the results. Including statistical analysis would greatly enhance the validity and clarity of the findings.

Response: We thank the reviewer for lacking this critical aspect.  In the revised manuscript we included statistical analysis for figures 4 and 5.

  1. TEM Image Clarity: The TEM images in Figure 6B do not clearly show the presence of AP. Could the authors provide more detail or clarification on this aspect?

Response: Unfortunately, the TEM operator didn’t provide better images that clearly show for the presence of AP. However, similar structures for adju-phos have been reported previously ( 10.1016/j.vaccine.2022.06.064 and 10.1016/j.biomaterials.2021.120960). 

  1. Neutralization Capability of Antibodies: While the study effectively measures antibody titers induced by the vaccines, it does not address the critical factor of how effectively these antibodies neutralize the virus. Evaluating the neutralizing capability of the antibodies would significantly strengthen the study's conclusions and relevance.

Response: We agree that the most relevant measurement would be the production of effective blocking antibodies. As our university does not have the ability to perform a true neutralization assay (requiring BSL3), we used the ability of the serum antibodies to block the interaction between the human ACE2 and Spike protein or isolated RBD domain (Figure 10) as a surrogate approximation of neutralizing potential.

Round 2

Reviewer 2 Report

Comments and Suggestions for Authors

The authors have answered all the questions satisfactory.

Reviewer 4 Report

Comments and Suggestions for Authors

Thanks for addressing all the questions. The quality of the manuscript is much improved.